# User Requirements Analysis of an Embodied Conversational Agent for Coaching Older Adults to Choose Active and Healthy Ageing Behaviors during the Transition to Retirement: A Cross-National User Centered Design Study

**DOI:** 10.3390/ijerph18189681

**Published:** 2021-09-14

**Authors:** Sara Santini, Vera Stara, Flavia Galassi, Alessandra Merizzi, Cornelia Schneider, Sabine Schwammer, Elske Stolte, Johannes Kropf

**Affiliations:** 1Centre for Socio-Economic Research on Aging, IRCCS INRCA-National Institute of Health and Science on Aging, Istituto di Ricovero e Cura a Carattere Scientifico Istituto Nazionale di Ricovero e Cura per Anziani, 60124 Ancona, Italy; s.santini2@inrca.it (S.S.); f.galassi@inrca.it (F.G.); a.merizzi@inrca.it (A.M.); 2Model of Care and New Technologies, IRCCS INRCA-National Institute of Health and Science on Aging, Istituto di Ricovero e Cura a Carattere Scientifico Istituto Nazionale di Ricovero e Cura per Anziani, 60124 Ancona, Italy; 3Institute of Computer Science, University of Applied Sciences Wiener Neustadt, 2700 Wiener Neustadt, Austria; Cornelia.Schneider@fhwn.ac.at; 4Nursing Programme, University of Applied Sciences Wiener Neustadt, 2700 Wiener Neustadt, Austria; sabine.schwammer@fhwn.ac.at; 5Gouden Dagen, 3721 Bilthoven, The Netherlands; elskestolte@hotmail.com; 6Austrian Institute of Technology, 1210 Wiener Neustadt, Austria; Johannes.Kropf@ait.ac.at

**Keywords:** embodied conversational agents, healthy aging, older workers, retirees, transition to retirement, users’ requirements, user centered design, virtual coach

## Abstract

Background: Retirement is recognized as a factor influencing the ageing process. Today, virtual health coaching systems can play a pivotal role in supporting older adults’ active and healthy ageing. This study wants to answer two research questions: (1) What are the user requirements of a virtual coach (VC) based on an Embodied Conversational Agent (ECA) for motivating older adults in transition to retirement to adopt a healthy lifestyle? (2) How could a VC address the active and healthy ageing dimensions, even during COVID-19 times? Methods: Two-wave focus-groups with 60 end-users aged 55 and over and 27 follow-up telephone interviews were carried out in Austria, Italy and the Netherlands in 2019–2020. Qualitative data were analysed by way of framework analysis. Results: End-users suggest the VC should motivate older workers and retirees to practice physical activity, maintain social contacts and emotional well-being. The ECA should be reactive, customizable, expressive, sympathetic, not directive nor patronizing, with a pleasant and motivating language. The COVID-19 outbreak increased the users’ need for functions boosting community relationships and promoting emotional well-being. Conclusions: the VC can address the active and healthy ageing paradigm by increasing the chances of doing low-cost healthy activities at any time and in any place.

## 1. Introduction

Between 1970 and 2017, the extension of life expectancy increased by 5.5 years in the Organization for Economic Co-operation and Development (OECD) countries [1]. This lifespan, however, entailed the onset of long-term care needs due to the increase in multi-morbidity [2,3,4], which significantly accelerates at older ages [5,6]. Therefore, the challenge of the twenty-first century lies in ensuring that the increased life expectancy can be lived in good health, meant as a multidimensional concept including physical health, emotional well-being, social inclusion and participation throughout one’s lifespan [7]. The promotion of health refers to two interrelated concepts: active ageing and healthy ageing. Active ageing [8,9], defined as “the process of optimizing opportunities for health, participation and security to enhance quality of life as people age” [10], is indeed the prerequisite for achieving healthy ageing, i.e., “the process of developing and maintaining the functional ability (i.e., people’s capabilities of being and doing what they have reason to value) that enables well-being in older age” [11]. Among the health status predictors at older age [12], the main predictors concern lifestyle: physical activity and psycho-social factors such as mental well-being, social networks, and social participation [13].

Moreover, retirement has also been recognized as a factor influencing the healthy ageing process, especially when it is experienced as a loss of life control and effort-reward mechanisms [14,15]. According to the life course perspective [16,17], healthy ageing partly depends on people’s behaviors during youth and adulthood, and partly on how individuals face transitions and changes over life. One of such changes is represented by the transition from work to retirement, which generally concerns people aged between 55 and 67 years according to the different pension systems and regulations that vary from country to country. This group of older adults also is in transition from the adulthood to the old age and so they do not fully fall in the traditional demographic and geriatric age and/or care classifications [18,19]. In fact, what defines them are not only age or long-term care needs, but even a multiple set of dimensions among which the level of participation in the society and the position in the labour market. The transition from work to retirement can embed both opportunities and challenges for different health realms. For example, after retirement, physical activity can be replaced by leisure time with activities more oriented to fun rather than to performance [20]. The decrease in physical activity could entail potential risks such as shortened life expectancy, cardiovascular disease, diabetes, and cancer [21]. Furthermore, although retirement is defined as a personal and not as a standardized process [22], this transition can entail complex psychological challenges for new retirees who are required to re-organize their daily schedule and relationship settings [23]. The impact of retirement on the older adults’ psychological and physical health seems to change over time, providing a protective effect in the first two years [24] which subsequently decreases and eventually disappears, thus creating room for a “disenchantment” phase that occurs when retirees realize that they have fewer resources and/or had unrealistic expectations on retirement. In the disenchantment phase, the initial protective effect of retirement on the health status experienced in the first two years of retirement decreases or disappears [25,26]. Another risk related to the transition to retirement could be due to the fact of receiving fewer cognitive stimuli, such as weaker and rarer meaningful social relationships, or a change in self-image, self-esteem and self-efficacy, leading to emotional discomfort, isolation [27], loneliness and depression [23]. Moreover, retirees may be concerned about their future, and especially about the deterioration of their health and about leaping into the unknown [28,29]. 

In light of the above, older adults transitioning from work to retirement should be considered as a group of older populations at risk of failing a healthy ageing process in the absence of focused policies and appropriate services and support measures. A possible way to counteract these missing actions could be using technology to improve and promote a healthy ageing process for people in transition to retirement. However, how to transfer the principles of healthy ageing to the technology field is still not clear. All the key components of a healthy and active ageing framework have captured the interest of various research fields in the area of innovative technologies and aging with the aim to find cost-effective solutions to support independent living and care provision [30,31] through digital coaching. Moreover, the efficacy of digital health coaching solutions is still little explored, despite there being a few examples [32] targeting people aged 65 and over [33] and also older workers aged between 55 and 67 years [34]. However, the few studies dealing with this issue give pretty strong evidence that web-based health promotion programs for older adult workers can improve their health conditions [35,36]. Moreover, the great potential of these technologies was evident during the COVID-19 outbreak, it is becoming crucial to continue doing physical exercise, and maintaining emotional balance and social contacts among older adults [37]. Furthermore, embodied conversational agents (ECAs) (hereafter also called “virtual coach”, “conversational agent” and “personal digital coach”) are seen as promising solutions aimed at coaching people for a healthy lifestyle [38]. They are currently an excellent example of natural, personalized, and more human-like human-machine interaction systems ranging from chatbots and 2D/3D to fully articulated embodied conversational agents engaged in ambient-assisted living environments [39,40]. Indeed, ECAs differ from other technologies (e.g., wearable devices) for having the same proprieties as humans involved in conversations such as the capacities to exchange verbal and non-verbal communication. The unprecedented advantage of having such rich style of communication and interaction modality lies in the fact that it offers human-like speech, facial expressions, hand gestures, and body stance and that it is supportive at any time and in any location owing to the use of applications on smartphones or tablets. These strengths are raising more and more attention in technology-based interventions, especially in health counselling, coaching, and self-monitoring [38,41].

Unfortunately, until now, the process leading to the design and content of embodied conversational agents is generally incomplete or missing, thereby restricting the discussion of lessons learnt from the work of others on this issue [38]. On the contrary, the user centered design (UCD) approach [34,42] can ensure that product solutions, especially when healthcare oriented, can respond to the target’s demand through a multidisciplinary, iterative multiple-step development process aimed at understanding the end-users’ needs, digital capabilities and perception of technology avoiding any stigmatization [42,43,44,45,46]. 

Therefore, to bridge the aforementioned gaps, this study aims to translate the active and healthy ageing dimensions into a virtual coaching system supporting older adults aged over 55, without long-term care needs, in aging well and healthily during their transition from work to retirement. 

To this purpose, the study addresses the following research questions: (1) What are the user requirements of a virtual coach (VC) based on an ECA for motivating older adults in their transition to retirement in adopting a healthy lifestyle? (2) How can a VC address the active and healthy ageing dimensions, especially during COVID-19 times? 

The first novelty of the study lies in considering a target that is often overlooked by research on the development of this kind of technical solution, i.e., people aged between 55 and 70 years, in transition from work to retirement. Moreover, this study wants to represent a new contribution for transferring the healthy ageing paradigm and the life course perspective to a virtual coaching system. Furthermore, it is worth mentioning that this study is one of the first studies, to our best knowledge, that monitored the end-users’ change of perspective concerning a virtual coach and an ECA throughout the COVID-19 outbreak. Finally, the collection of the end-users’ inputs, especially on the ECA, may enrich the knowledge on this innovative solution and pave the way towards its full personalization.

## 2. Materials and Methods

### 2.1. Study Design

This study is based on data collected during the European project AgeWell funded by the Active and Assisted Living Programme, with the aim to develop a virtual coaching system running on Android. The VC is connected to the other components via message queuing telemetry transport (MQTT) standard messaging protocol for the Internet of things (IoT): the ECA, the physical activity platform, the health platform, and the content. The whole process of the VC design followed the UCD [47] approach, whose four basic activities were performed: (a) user groups are specified and the context of use is described (Activity 1: understand and specify the context of use); (b) a set of specific requirements is defined in order to create a degree of fit between device and user (Activity 2: specify the user requirements); (c) the design prototypes are produced on the basis of these specifications and are presented to the user in the form of user testing (Activity 3: produce design solutions to meet requirements); (d) once feedback has been received, the process begins again until all user requirements have been met (Activity 4: evaluation). 

This paper shares the lessons learnt from the development of Activity 1 and 2, by providing a longitudinal and cross-national analysis of the data collected in Austria, Italy and the Netherlands between Spring 2019 and Winter 2021 as reported in Figure 1.

The UCD Activities 1 and 2 were performed by adopting a qualitative approach based on two-wave cross-national focus-groups and a telephone follow-up study with end-users, carried out during the COVID-19 outbreak, in Austria, Italy and the Netherlands. 

The two focus-groups, conducted in Spring and Autumn 2019, were aimed at exploring the user requirements of the initial, and then of the advanced concept idea of a VC based on an ECA designed for promoting the health and well-being of retirees and older workers close to retirement. 

The follow-up study was aimed at understanding how and to what extent the ideas of the end-users on the solution had been influenced by the COVID-19 outbreak and lockdowns, and if the system could have been helpful for mitigating the effects of the pandemic restrictions (e.g., physical distancing and “stay at home” governmental measures) on the well-being of older workers and retirees, especially under the three dimensions identified by the end-users and addressed by the system, i.e., physical health, emotional well-being and social inclusion.

### 2.2. Participants’ Inclusion Criteria and Recruitment Strategy

The focus-group study targeted older workers, retirees and the family members and/or ex-colleagues of both. Older workers were included if aged 55 years or over, having at least three years before retirement and working as “white collar”, i.e., sitting, standing, walking or moderate working activity. In fact, since the literature shows that physical leisure-time activities decrease with retirement from a physically demanding job [48], older workers doing physical demanding jobs were excluded from the study in order not to negatively influence the participants’ attitude in thinking about system functions related to physical exercise during transition to retirement.

Retirees had to be 55 or over, retired for three years and in good health, i.e., not having any severe physical or cognitive impairment, which could prevent them from thinking about a system aimed at motivating older adults to have an active and healthy lifestyle. 

The only inclusion criterion for colleagues and family members of retirees was being in a close relationship with the latter. Colleagues and family members were included in the study as persons in close relationship with the participating retirees or older workers. They were supposed to know habits, expectations, fears and plans for retirement of the retirees and older workers involved in the study so they could easily empathize with them and provide a further perspective on their transition from work to retirement.

The inclusion of different perspectives from the three types of participants, i.e., older workers, retirees and respective colleagues, ensured the study’s validity [49].

In Austria and Italy, participants in the focus-groups were recruited mainly within the research organizations carrying out the study, their network, human resource offices and local companies. In the Netherlands, the NGO carrying out the study sent an invitation letter to all care center users and published advertisements via press and social networks, e.g., Facebook. It is worth mentioning that the study organizations are large bodies or part of public entities and NGOs that employ or are connected to hundreds of people with different professional profiles, e.g., administrative staff, doctors, nurses, sociologists, psychologists, social workers, engineers. The participants in the study did not know either the focus-group topic or the moderators, and thus their opinions were fully independent and simply the expression of their own thoughts. 

In the three study countries, researchers phoned the potential participants, checked they met the inclusion criteria, explained the study’s purpose and methods, and proposed that they take part in it. Individuals matching the inclusion criteria and available to take part in the study were provided with and asked to sign a written informed consent to data treatment, in accordance with the GDPR 2018 and the national legislations on privacy and data protection. In Italy and in the Netherlands, given the observational non-clinical nature of the study, the Ethic Committee’s approval was not mandatory, whilst it was mandatory in Austria, thus the study was approved by the Ethics Committee in charge.

In January 2021, researchers newly contacted the end-users already involved in the focus-groups by phone and asked for their availability to answer six open-ended questions by phone on the VC and on its usefulness during the COVID-19 outbreak. Only 27 people accepted to take part in the follow-up study. Indeed, many end-users were not interested in thinking about a system that was still theoretical and that they had never used, especially in times when people were concentrated in putting in place all the possible actions aimed at avoiding the virus infection for themselves and their loved ones.

### 2.3. Data Collection Methodology and Tools 

The focus-groups were carried out in two rounds and three discussions were managed per round in each country with the three types of participants. The discussions were conducted by a researcher playing the role of moderator, whilst another researcher observed the discussions, took notes and timings. The moderator showed Microsoft Power Point slides to introduce the materials described above.

The first focus-group round was aimed at identifying end-users’ needs and expectations on health and well-being in the specific phase of transition from work to retirement and at exploring their reactions and toward the ECA.

In this first wave, a set of scenarios, definitions and storyboards (Figure 2) were shared with the participants for triggering the conversation. Storyboards were created using an online tool for comic creation. In total, comic strips for six use cases were created, each containing 4–9 images. In all comics, the same visual with an avatar on a smart phone was used. For the backgrounds, familiar and appropriate scenarios were chosen (e.g., bedroom, living room, park, café). 

The topic-guide for the first wave focus-groups discussions (Appendix A) was developed along four different areas: (a) feelings and needs on retirement (retirement representations); (b) concept: feelings and needs regarding the coaching system; (c) functions and services of the VC; (d) appearance of the ECA and usage of the VC.

The second round of focus-groups was aimed at capturing the end-users’ ideas on the VC basic functions and requirements, so attempting to ensure that it could match the target’s demand for acceptance, usability and reliability. End-users were introduced to the contents of the app running the VC by means of mock-ups (Figure 2), translated from English (the language used by the international research consortium) to German, Dutch and Italian languages. Then, the participants discussed which functional areas they would have liked to have been coached on by choosing amongst the three well-being realms that they had identified, i.e., physical activity, social well-being and emotional support. Furthermore, end-users’ preference for the various functions and services of the VC was gauged and suggestions for improvement were collected. 

Thus, the second wave topic-guide, mirroring the above aspects, covered four issues: (a) frequency of use of the VC in daily life; (b) design and usability of the VC; (c) tasks, functions and usefulness of the VC; (d) appearance of the ECA (Appendix A).

The follow-up study’s topic-guide (Appendix A) was developed through six questions asking about the impact of the pandemic on respondents’ health and well-being and the potential use of the system in mitigating such an impact, especially in the three dimensions addressed by the system, i.e., physical health, emotional well-being and social inclusion.

### 2.4. Data Analysis Methodology

All textual data, i.e., data from focus-group discussions, and telephone semi-structured interviews with the end-users were mostly recorded digitally and transcribed verbatim. When this was not possible, such as for three respondents who did not accept the recording of the conversation, researchers transcribed the contents of the interviews as literally as possible. 

Although the interviews were recorded and transcribed or straight transcribed, the contents were analyzed with the support of MAXQDA Plus software [50], by using the framework analysis method [51,52,53,54]. Researchers familiarized themselves with the texts through immersion in the raw data. The data obtained from the texts were divided into chunks and associated with codes systematized into a tree-chart. The codes were combined under main themes, and the latter were identified once the consistency within different codes under the same theme had been assessed. Repeated patterns/themes throughout the data set were identified and a code was associated to every chunk of text. In line with the literature [55,56], the content areas expressing similar concepts were grouped into mutually exclusive categories that were associated to codes. When there was disagreement on an issue within the national groups, it was underlined and reported under a different code.

As a second step, two or more codes were combined, and different codes were sorted into themes (all themes and codes are shown in the Multimedia Appendix A). Then, the data were sifted and this activity led to sorting quotes and making comparisons between them [57]. The data were rearranged according to the appropriate part of the thematic framework to which they related. Therefore, the analysis started deductively from the aims and objectives of the study embedded in the topic-guide, but also reflected the original observations of the respondents accordingly with an inductively approach. The final step was the interpretation of the data through associations between themes and within the different cases. The interpretation of the data was the result of an iterative work carried out by a multidisciplinary team consisting of psychologists, sociologists, experts of UCD, engineers and technicians. The parallel and independent data analysis by three researchers minimized the research bias [58,59,60,61,62]. 

## 3. Results

### 3.1. Participants’ Description

In the first wave of focus-groups, 45 people participated (21 males and 24 females), out of whom 18 were older workers, 14 retirees and 13 colleagues or relatives of retirees. The mean age was between 59 and 65, 6 years across the three countries. In the second wave, 60 individuals were reached, consisting of 29 males and 31 females. In Italy and in Austria, the participants were the same subjects involved in the first wave even if in Austria, there were 6 drop-outs; in the Netherlands, during the second wave, 21 individuals joined the group that had already participated in the first wave of the focus-group (Table 1).

It is worth underlining that all participants of the study in the three countries were digitally literate: they already had smartphones and tablets, were familiar with using applications for communicating with friends, making online payments and monitoring physical activity as well as surfing the internet for seeking information and news on local social and cultural events. Someone was also used to wearing a smartwatch for monitoring sport performance and health parameters.

In total, 27 individuals were involved with the telephone follow-up interviews in the three study countries (Table 2).

### 3.2. Overview of User Requirements as Determinants of Use S of a VC 

The analysis of the contents emerged from the focus-group discussions in the three study countries identified five user requirements: (1) general concept (i.e., objective and meaning) and perceived usefulness; (2) intervention areas and functions; (3) frequency of use; (4) users’ perceived risks; (5) design.

Since behavioral intention is considered a key predictor of technology use behaviors [43,44,45,46], the early detection of the user requirements, accordingly to the UCD, can optimize the match of the VC design and the users’ expectations. This allows increasing the chances that older workers and retirees use the system because they find it really useful in order to have a meaningful and healthy ageing, especially during the transition from work to retirement. Such user requirements can be therefore considered as determinants of older workers and retirees’ behavioral intention to use the VC with the aim to maintain health and well-being.

Figure 3 illustrates the findings of the focus-groups concerning the five VC user requirements identified by the study, that were framed in five boxes. The box named intervention areas and functions of the VC includes the active and healthy ageing dimensions. The arrows represent the influence of the user requirement on the intention of older adults to use the VC as well as the effect on the system use behavior that impacts on the behavioral intention itself. 

For every user requirement, several themes were identified in every study country, answering the two study research questions, e.g., “end-users’ opinions on the VC functions”, that are fully reported in Table 3. 

Such themes represent the main results of the study that are extensively reported in this section and that might be useful inputs for technology developers to better understand the preferences of older adults in transition to retirement concerning the virtual coaching system and the ECA.

The quotations extracted from the answers of the participants in the study are reported in this section in support of the themes identified by the analysis and answering the research questions. The quotations represent examples of the general orientation of the group of respondents or, conversely, the exceptions to it. Since the study included three types of participants, i.e., older workers, retirees and colleagues/family members—in most cases the authors of the quotations are specified. When the participants agree on a topic, the type of respondent is not specified and the quotations are intended to refer to all the respondents. Every quotation is followed by basic information concerning the respondents, i.e., country, gender and age (e.g., A, female, 58). Such information is not available for the Dutch participants in the focus-groups, because the respondents did not give their consent to disclose the data. However, it is available for the ones participating in the telephone follow-ups, who gave their consent on that occasion.

In the following sub-paragraphs, the user requirements are reported in detail, answering the two research questions.

### 3.3. What Are the User Requirements of a VC Based on an ECA for Motivating Older Adults in Transition to Retirement to Adopt a Healthy Lifestyle?

#### 3.3.1. End-Users’ Feelings on the Concept and Usefulness of the Coaching System

In the first focus-group, participants discussed the general coaching system concept by answering Questions 3–8 of the topic-guide (Appendix A).

In the Austrian groups, at first some participants totally refused its use, while others reported to be open to its use as a system for managing health issues and counteracting loneliness and frailty: *“Maybe in 30 years I would be happy to have such a thing, but in my current situation I do not want this […] I do not need it now*” [A, Female, retiree, 66]. 

Similarly, most of the Dutch respondents did not see the coaching system usefulness, as expressed by the following quotation: *“I myself don’t need it at this point of my life!”* [Dutch privacy rules applied].

In Italy, retired people as well as older employees did not see the usefulness of a VC because, in their opinion, this kind of technology can be useful just for people with physical and cognitive impairment. Therefore, as they feel, and actually are, in a good health condition, they initially did not find a virtual coach a useful device: *“Today I don’t need these things, but maybe in 10 years I would think otherwise”* [IT, female, 62]. The prevalent sentiment was mistrust towards a technology which “wants to say what I have to do”, especially because Italian retirees viewed retirement as a chance to be free and masters of their own time: *“In my opinion, retirement also means to respect one’s time. If I have a technology that pounds me all day, I no longer see respect for the pensioner. I don’t want to have schedules or programs anymore and I don’t want technology to force me to do what I have to do. I see it as a lack of respect for my timing and freedom. Moreover, what I want to do changes from day to day, so it would be very difficult even to personalize the virtual coach’s inputs”* [IT, female, 64].

The main difference at cross-national level concerning the concept and beneficiaries of the general system lies in the fact that the Austrian and the Dutch participants suggested the use of the VC mainly at individual level for improving health and socialization, while the Italian groups underlined its usage also at company level: *“The virtual coach may be used by companies to facilitate the transition from work to retirement… there should be a more structured intervention by companies, to help the workers to live this passage peacefully. The use of the virtual coach should be a business service to offer to workers. From this point of view, the workers close to the pension can turn to the system and ask for information about retirement”* [IT, male, 52].

#### 3.3.2. End-Users’ Opinion on the Intervention Areas of the Virtual Coach

Three main intervention areas were identified by the end-users for the virtual coach: (a) physical activity; (b) social relationships and (c) emotional well-being. 

When we look at national specificities, the Austrian participants, especially retired respondents, stressed the intervention of the VC for motivating retirees to practice more physical activity and for providing cognitive stimuli: *“It would remind me to do exercise for example by the pedometer” [A, male, 57]*. Conversely, older workers underlined the usefulness of interventions boosting socialization: *“If it is linked to a platform I could also find other retirees when I will be retired with the same interests and free time”* [Dutch privacy rules applied]. 

The Italian retirees who were more attached to work and missed social contacts and cultural inputs, were more open to the idea of using a VC, especially for boosting socialization, and stimulating interests and brain activities. Among the Italian participants, the necessity arose for a coach that can meet the numerous changes which people have to face during the transition from work to retirement: *“In the transition from work to retirement there is an economic change, a change in health, in legislation, in educational initiatives and in facilitations. People often don’t know a lot of things because they never got the necessary information before retiring. They lose so many opportunities because they do not know about them until they find out, so it may be useful to inform them in the transition phase between work and retirement”* [IT, female, 60].

In the Netherlands, positive feelings mostly had to do with practical help, either financial planning or planning of concrete activities and health screenings. Coaching on healthy living was also seen as a good option by most. Many saw the virtual coach as a useful option for older people with physical limitations and/or living alone: *“Maybe if I were alone I would like it. Now I have my husband”* [Dutch privacy rules applied]. 

#### 3.3.3. End-Users’ Opinions on the Frequency of Use of the Virtual Coach

In the second focus-group, participants were asked frequency of use of the VC (Questions 1-4 of the first-round topic-guide) (Appendix A). 

Participants from the three study countries agreed that the frequency of use depends on what the coach is used for. In fact, most participants thought that they could use the coach every day if it contained appropriate and useful activities and addressed their needs. 

In Austria, two participants stated that they were keen to use the system every day and multiple times a day: *“Surely multiple times daily—but I am very technophile”* [A, male, 63]. Conversely, others thought to use it when needed and depending on their mood: *“I could imagine using it for a few weeks only when I feel bad and then I do not need it anymore”* [A, female, 63].

In Italy, some interviewees underlined that the usefulness of the virtual coach could vary depending on the user’s family, social relationships and commitments as depicted by the following quotation: “*Obviously the variability of the experience of the transition to retirement is enormous, there are those who have their own hobbies and do not need anything else, there are others who, on the other hand, live through the transition badly and run the risk of shutting down. So an active person doesn’t need this avatar, but a single person needs something more than what I see the virtual coach is currently offering”* [IT, female, 66].

In the opinion of some of the participants from the Netherlands, the use of the coach depends on the free time available. Especially retired people could need support for better time management and exploitation, given the great amount of free time that retirees may suddenly have at their disposal: *“Owning your own time seems nice, but I think there is a risk of feeling ‘lost’ or ‘without purpose’”* [Dutch privacy rules applied]. 

All participants from the three study countries said they prefer using the system on a portable device and preferably on a smartphone because it is lighter and less bulky when you want to practice the tasks proposed by the virtual coach outdoors (e.g., walking, mindfulness and activities with friends).

#### 3.3.4. End-Users’ Opinions on the Risks of the Virtual Coach

The participants in the three countries agreed on three risks identified coming from the use of the VC: (i) stigmatization; (ii) privacy violation; (iii) substitution of human beings’ cognitive and decision-making capabilities. The virtual coach can be stigmatizing if it is not tailored to the real needs of older adults in good health and still active: *“Needing a system like this means being old and not self-sufficient. Thinking I need it, makes me feel even older. Since I don’t feel old I don’t feel the need to have a robot”* [IT, female, 64].

Moreover, it could violate the individuals’ privacy in the absence of a proper system for data protection: *“Privacy is a big issue. It should not share any of my information with anyone. No commercial links either”* [Dutch privacy rules applied]. An Austrian participant clearly stated: *“I don’t want a system that is permanently controlling me. I would become crazy”* [A, male, 57].

Furthermore, in the opinion of many respondents, the virtual coach might start to replace human being’s decision-making capabilities if it is too proactive and intrusive: *“In my opinion, there could be a violation of decision-making capacity”* [IT, female, 66].

#### 3.3.5. Design of the ECA

During the second focus-group, the user requirements of the ECA were proofed through the Questions 5–7 of the second focus-groups topic-guide (Appendix A). The participants mainly focused on: communication (voice and language), appearance and interface design. 

In Austria and in the Netherlands, the communication between the user and the system was an important issue. It was considered as positive to be able to talk/have voice control and to set the voice personally: *“What is important to me is that the embodied conversational agent can communicate; that I can give verbal orders and it understands me”* [A, male, 66]. Several older workers suggested the ECA should be controlled and regulated by the users’ voice as to whether the device starts speaking or just reacts, or to set a time slot for allowing the system to speak. However, most participants in the three countries thought that the ECA should be reactive and not proactive, in order not to be too intrusive, and thus it should suggest activities but after the person’s request in order to respect his/her desire: *“In my opinion, retirement is also respect for one’s time. If I have a technology that pounds me all day, I no longer see respect for the pensioner. I don’t want to have schedules or programs anymore and I don’t want technology to force me to do what I have to do”* [IT, female, 62].

The debate around the avatar’s voice was pretty lively. All participants in the three countries suggested to make the voice tone of the ECA more human and more melodic: *“She has a commanding tone, like a senior teacher with a raised index finger, like a control function”* [A, female, 40]. 

They also underlined the importance of an appropriate language used by the ECA: it should not be too directive, non-patronizing but, at the same time, convincing, motivating and respectful of the users’ vulnerabilities. 

Concerning appearance, in the three countries, the ECA was seen as too formal, serious, rigid, not expressive and not sympathetic: *“Just not playing the teacher…that would put many people off, they get stubborn then”* [A, female, 40]; *“I don’t like a personified avatar, I prefer something more like a game”* [IT, Male, retiree, 67], and *“This avatar is grey, 70s and very boring”* [IT, female, 37]. The Dutch participants especially underlined that they felt uncomfortable with the avatar because it pretended to be human: *“An avatar is fake too. You see it is not a human being and the way it moves is not human”* [Dutch privacy rules applied].

In order to overcome this limit, the end-users suggested the customization of the virtual coach, e.g., the possibility of choosing the avatar’s gender and age, and of changing the facial expression: *“Remove the jacket, for example, if you tell me to do physical activity, you are dressed in gymnastics, if you help me to cook, you are dressed as a chef, if you remind me of the check-up, you are dressed in a white coat”* and *“Let her smile and change expression with respect to the message conveyed….Improve the relational capacity of communication. Depending on the message conveyed it must be different (assertive, kind, proactive)”* [IT, male, 66]

Concerning the interface design, Italian participants also recommended to make it possible to have icons that give feedback to the users in order to motivate them to improve their performance in the different health dimensions covered, as depicted by the following quotation from a retiree’s colleague: *“When the interface has to give me a positive feedback, the icon can also pulse. It would be a double reinforcement that is very important in the case of coaching”* [IT, female, 37].

Moreover, in Italy there was unanimous opinion about the necessity of avoiding a purple background for the avatar and customizing icons as much as possible: *“The graphical representation of the information on the screen should be customizable. I should decide if I prefer a compact iconic menu or a more textual list”* [IT, female, 55]. 

Conversely, the Austrian end-users preferred a bright and changeable background, whilst the Dutch end-users liked the purple background.

The participants unanimously underlined the importance to make the text field and colours appealing: *“The white box with the black border and the text field remind me of the warnings of a cigarette box”* [A, male, 41].

### 3.4. How Can a Virtual Coach System Address The Active and Healthy Ageing Dimensions, Even at COVID-19 Times?

In the second focus-group wave, the participants detailed the VC functions according to the three intervention areas agreed upon during the first wave, by replying to and discussing Questions 12–19 of the second wave topic-guide (Appendix A). 

These findings answered the first part of the research question (i.e., “How can a virtual coach system address the active and healthy ageing dimension […]?”), while those collected through the telephone follow-up interviews answered the second part of the research question (i.e., “[…] especially during the COVID-19 outbreak?”). 

#### 3.4.1. The VC Functions Promoting Physical Health

In the physical health area, in the Austrian participants’ opinion, the most important function of the VC should be to monitor the end-users’ health in combination with other apps monitoring health parameters both for older workers and retirees: *“It would be useful if I could combine the service with my pulse and blood pressure. As an older person exposed to many health risks, I would like to have an overview of my physical status”* [A, male, 58]. The VC should be customizable especially for people with special chronic diseases, e.g., diabetes: *“One could also make such a health app if one has a special problem, diabetes for example, combine health advice or health information”* [A, female, 40].

The Italian participants stressed the importance of preventative health by reminding and planning health and screening appointments, especially for older workers who are busier and may forget their medical appointments: *“For me it would be very useful if the ECA can remind us (i.e., older adults/retirees) of fulfilments over time,* e.g., *blood test or examinations that are important at a certain age at least once a year. Health comes first!”* [IT, female, 66].

The Dutch end-users underlined the need for stimulating the older adults’ brain activity: *“A knowledge quiz would be interesting, with single or multiple-choice answers and several players, followed by scoring. Yeah, like the million-dollar show app”* [Dutch privacy rules applied].

#### 3.4.2. The VC Functions Promoting Emotional Well-Being

In the participants’ opinion, especially the Austrian and the Dutch, emotional support was deemed a very important service that could be delivered by the VC, because it was very much associated with meaningful life, purpose in life and stress reduction, important themes during retirement. Therefore, the VC may prevent low mood and depression that can be experienced by retirees and older workers. Nonetheless, its feasibility was considered doubtful, because of the sensitiveness of the issue which, if treated incorrectly, could have an adverse effect on frail people (for example on subjects suffering from depression): *“The ‘emotional support’ service is the most difficult service. I have friends who suffer from depression and the worst thing to imagine is the idea of being asked every morning: how are you today? […] I think there should be an attentive way, because if the avatar arrives every day with this question, it helps you jump out of the window”* [A, male, 67].

Moreover, a Dutch participant underlined that the system may help the overall well-being of older adults’ in transition to retirement, by including functions supporting the emotional status: *“Balance in life and a feeling of meaningfulness are important to me and the virtual coach should help me to reach this balance”* [Dutch privacy rules applied].

Converely, the Italian participants did not appreciate and trust the function promoting emotional well-being: “The function for providing emotional support is quite weak. I think it is not sufficient to write a post-it to be happier. This representation is too much simple […] To better the mood of people it is more useful ask them which is their favourite music or book and play or read it” [IT, female, 67].

#### 3.4.3. The VC Functions Promoting Social Relationships and Participation

In Austria, the VC was considered an attractive means for enhancing the older workers’ and retirees’ social life and motivating them to connect with others and participate in the community: “*It should show the possibility to connect with other people and tell me if there are dancing groups or other retirees who want to connect”* [A, female, 66].

Similarly, the Italian end-users stressed the importance of the VC for connecting to other retirees with similar interests: “It could be nice the VC help share objectives and goals, especially in cultural activities; the avatar could inform the user about cultural events” [IT, female, 62]. 

Conversely, the Dutch end-users did not consider socialization a basic function and they stressed the efficacy of stimulating people by means of competition: *“I don’t think I would like to share with others. It is my personal coach […] Working with kudos or some kind of competition with others on the same platform can stimulate (physical) activity”* [Dutch privacy rules applied].

#### 3.4.4. The Influence of the COVID-19 Outbreak on the End-Users’ Opinions about the Virtual Coach’s Coach Functions

The participants in the telephone interviews carried out during the second COVID-19 wave outbreak suggested the increase in the number of activities and reminders motivating the older adults to re-start going out as soon as possible and in compliance with the restrictions in order to reduce the habit of staying at home when it no longer necessary. *“I think it is important to encourage older people in retirement to go out and walk or to stay in the open air, even during the lockdown, maybe by referring to the daily weather forecast or to the fact that one gets bored by staying at home too long. Moreover, the app should push older adults to practice physical activity at home”* (IT, female, 62).

In the end-users’ opinion, the VC may also motivate retirees and older workers in smart working during the lockdown, to do indoor physical and yoga exercises, for instance by using video tutorials and online work out classes: A retired person in the Netherlands said: *“Going out into the fresh air is difficult when everything is forbidden. There are certain programs on TV in the morning, such as gymnastics exercises which are tailored to older people; something similar could also be offered in the app to keep you physically active”* (NL, male, 62).

Reminders about physical exercise were especially recommended: “During the lockdown the VC may stimulate you to make walking appointments with people you know or cycling appointments. Give the tip to use the exercise equipment in public parks” (A, female, 60), and “I think the virtual coach should have helped me remember to move. In this period when you are even less active at home, between smart working and it being almost impossible to go out, it is important to exercise at the right times to stay healthy” (IT, female, 63). 

Concerning emotional and social well-being, the most important issue was the addition of appropriate information on movement restriction measures, forbidden places and areas where it was possible to go for a walk, in order to help the users’ peace of mind and ensure that they could enjoy outdoor physical activity without the fear of getting a fine from the municipal police or of being infected. Other suggestions concerned the proposal of mindfulness exercises (especially within the Austrian groups of end-users), virtual visits to gardens and museums, online courses and games also for increasing remote socialization. Someone suggested to add a function for e-learning: *“It could help with new hobbies, for instance, to follow an online first aid course; I did this during the first wave and it was quite nice. I got a diploma for it. Or offer/suggest puzzles. Also help focus on the positive and what you can do. Make sure you have a purpose”* (NL, female, 68).

The interviewees underlined the need of a VC that can motivate older adults to stay in contact with friends and relatives during the pandemic: *“A lookalike of social media such as FB. Make it possible to find people with similar interests by specifying a hobby, for instance. Low threshold access to ‘clubs’, for instance, where you can read messages to get an impression without having to join immediately. Add ability to react to each other”* (NL, male, retired, 61). The latter suggestion was echoed by end-users in Italy suggesting to improve the construction of a community even by creating groups of users based on common interests (e.g., history, bricolage, etc.).

Moreoer, in the Dutch participants’ opinion, during the pandemic, the system could have helped people ask for social and health support if needed: *“The coach can help ask questions about contacts. Encourage people to ask for help if they need it, for instance in buying groceries. Don’t be ashamed. Ask every other day how someone is doing and if they have had contacts with someone else”* (NL, male, retired, 66).

Furthermore, the Italian end-users confirmed the need for connecting the ECA inputs to the real activities and events available at local level: *“Perhaps after so many months of virtual contacts, stimulating (when possible) people to have a social life is important. Motivational messages in this sense could be useful. In particular, considering the fact that one of the side-effects of the pandemic period will probably be a certain mistrust of the other”* (IT, female, 58).

## 4. Discussion

The aim of this study was to collect end-users’ inputs for the design of a VC system based on an ECA that addresses the active and healthy ageing needs of older workers and retirees.

The first novelty of the study lies in considering people aged between 55 and 70 years, in transition from work to retirement, as users of a VC based on an ECA. In fact, this target is often overlooked by research on the development of this kind of technical solution [24]. The involvement of end-users in different stages on the way towards retirement (i.e., older workers, retirees and colleagues/relatives) provided a multifaceted perspective on the retirement experience, as well as on the acceptability of the system and user requirements. Moreover, this multi-perspective approach led to a multidisciplinary analysis and comprehensive interpretation of the data that considered social representations of ageing and retirement and cultural pattern for the definition of the user requirements. Indeed, this approach can represent the only one that is able to describe the interactions between three very complex phenomena: ageing process, retirement and use of an ECA agent.

Furthermore, this study represents a new contribution for transferring the healthy ageing paradigm and the life course perspective [8,9,10,11] to a virtual coaching system. It is also one of the first studies that monitored the end-users’ change of perspective concerning a virtual coach and an ECA throughout the COVID-19 outbreak.

Finally, the collection of the end-users’ inputs on the ECA had enriched the knowledge on this innovative solution and paved the way towards its full personalization. Indeed, the ECA uses the same communication channel as human beings, verbal and non-verbal, i.e., voice and facial expressions, so it can stimulate a natural relationship between the user and technology [38,39,40,41,63].

The older workers and retirees involved in the study reported very different experiences of pre- and post-retirement that is confirmed to be viewed as a personal, non-standardized, and fluctuating experience affecting many life realms (e.g., social contacts, finances, levels of physical and brain activity) in different ways over time, and thus potentially influencing the individuals’ overall health and well-being [21,22,23,24,25,26]. 

Despite the personal perspectives on retirement transition, the end-users in the three study countries agreed that retirement is a period of life characterized by freedom, independence and decision-making capabilities. Some end-users, therefore, defended the freedom of being masters of their time after retirement without the need for suggestions from a digital assistant. On the other hand, others were more open to the use of the system, finding it useful for planning and exploiting the free time gained with retirement. 

The abovementioned observations lead us to think that the participants’ initial reluctance towards the use of the VC might depend on three main factors: the representation of retirement as the beginning of ageing; the fear of being stigmatized as “old”, due to the use of the coaching system; the fear of losing independence and decision-making capabilities. Nevertheless, the end-users recognized the potential of the virtual coaching system in motivating older adults to have healthy behaviors during the retirement transition [34,35,36]. They identified the following basic system functions: (i) motivating older workers and retirees to do physical activity and monitor their health conditions; (ii) stimulating retirees’ brain activities; (iii) providing older workers and retirees with emotional support, especially when older workers are leaving the work place; (iv) informing users on retirement legislations and providing advice on financial issues; (v) motivating older workers and retirees to be connected to other people; (vi) promoting community building also by informing users of initiatives carried out (in person as well as digitally, thus remotely) at a local level by municipalities and private providers.

### 4.1. What Are the User Requirements of a VC Based on an ECA for Motivating Older Adults in Transition to Retirement to Adopt a Healthy Lifestyle?

The study advances the current knowledge on the virtual health coach systems based on ECAs [38,41] by providing qualitative evaluations of the features that users like and dislike. In the study, the participants’ opinion was that the ECA should mirror the personality of a coach. Moreover, it should have an empathic verbal and nonverbal behavior, give motivational, supportive and positive messages, and adopt a language that is not too directive nor patronizing. 

These qualitative evaluations are significant in order to identify structural, physical, and psychological barriers that could affect the use of virtual agents. Such high-quality evidence is a fundamental key to assess how ECAs could or could not work in health care, or which feature improvements could enhance their usefulness [63]. Furthermore, there was a general agreement on the need for a full customization and personalization, e.g., clothing, gender, age as well as information, communication style or motivational cues. 

The participants in this study clearly confirmed a well-known principle that a ‘one size fits all’ approach [64] is not suitable. Indeed, the need is to adapt content and functionalities to the aims, behaviors, preferences, context, and lifestyle of the intended user. This means “changing the system functionalities, interface, content or distinctiveness to increase its personal relevance to an individual or a category of individuals” [65]. For example, personalization is primarily used for tailoring content through feedback, daily health reports, alerts, warnings, and recommendations. Nevertheless, this recognized added value is not yet considered a distinctive design factor [66].

Consistently with a systematic review evaluating design features of ECAs in eHealth, older populations seem to prefer images of young, female agents over male ones, and characteristics such as friendliness, expertise, reliability, involvement and authority [42]. 

The design and development of an ECA based on the UCD approach is a valuable but also challenging issue. Matching needs with technology remains a great challenge since, despite the value of each need, there are time and effort restrictions that constrain research and development studies. Nevertheless, the personalization of ECAs will be crucial for future development in these fields [66]. In the literature, a couple of effects of the interaction with ECAs have been reported which need to be considered in the development, e.g., the uncanny valley effect describes the hypothesized relationship between the degree of an object’s resemblance to a human being and the emotional response to such object [67,68]. In our case, imperfect realizations may provoke uncanny or strange feelings in observers. For the development this means the more human-like the avatar looks, the less imperfections are accepted by the user. Therefore, more effort needs to be put into development and implementation (e.g., lip synchronization, graphical details or speech synthesis). Another challenge is the increased expectations of intelligence from the avatar, the more human-like it appears to be. In particular, when speech recognition and speech synthesis is used, a very realistic avatar is expected to be able to interact similar to a real human. Hence, it is more advisable to make shortcomings obvious (e.g., by creating an Avatar which looks more artificial, similar to a pet, etc.) and to avoid raising high expectations which cannot be maintained and may lead to feelings of frustration in the user.

### 4.2. How Can a Virtual Coach System Address The Active and Healthy Ageing Dimensions, Even at COVID-19 Time?

This study provided evidence that a virtual coaching system can address the determinants of active and healthy aging by enabling older workers and retirees to carry out healthy physical and social activities whenever and wherever they want and at low costs, [10,11,12,13] e.g., doing physical exercise, taking personal time, meditating, and being informed of local events and of places where to meet friends.

The use of a virtual health coach can be especially useful in the transition from work to retirement, which is considered a determinant of health in later life [14,15], as it can mitigate the negative effects of retirement, such as the decrease in physical activity, cognitive stimuli and social interactions [20,21,22,23,24,25,26,27,28,29]. Moreover, in the end-users’ perspective, the VC can also influence the ageing well process by providing tailored information and advice on retirement transition, e.g., on legal and financial issues. 

The COVID-19 outbreak made those end-users who were suspicious about the virtual coach more open to the idea of using it. The pandemic led many respondents to prioritize functions that can enhance physical activity, emotional well-being (e.g., mindfulness exercises) and socialization, possibly mirroring the routine activities that the users missed most during the lockdown [27]. The end-users stressed that, during the COVID-19 pandemic, the virtual coach might have been useful for covering the three healthy ageing dimensions identified by the participants involved in the study. 

Under the realm of physical health, the virtual coach could have encouraged older workers who were smart-working to practice indoor physical exercises, and motivate retirees to re-start going out as soon as it was possible. Moreover, the system could have been useful for providing information on the pandemic and supporting people in asking for help from social and health care professionals. Furthermore, the system could provide users, and especially retirees considered more at risk of social isolation, with mindfulness exercises, virtual visits to gardens and museums, online courses and games for socializing.

This suggests that the COVID-19 pandemic might have accelerated the acceptance of the virtual coach system and that it could be seen as useful by older adults for maintaining physical health, emotional well-being and meaningful relationships, even when any kind of adverse condition that would force people to stay at home may occur, e.g., a flu, a snowfall or high pollution levels.

### 4.3. Suggestions for Future Design and Research

The end-users’ initial reluctance to use the virtual coach suggests the need for a system that must not be stigmatizing in order to be fully acceptable [43,44,45]. To this purpose, firstly, the functions of the system should be related to health aspects and not to the end-users’ age, so as to avoid any possible association with ageism-oriented perspectives and stereotypes. Secondly, it should be personalized and be able to make a comprehensive user profile by asking for relevant data at first use, and again periodically for changes to user preferences and personal interests (e.g., sport, music, cinema, reading, etc.). In this way, the activities proposed by the virtual coach can meet the older workers and retirees’ interests, stimulate and motivate them to do such activities, and finally gratify and provide positive reinforcement to achievements. 

The older adults taking part in the study seem to continuously adapt to the changes occurring in daily life, tasks, social and family relationships, all affecting their physical and emotional well-being. In light of these inputs, a virtual coach that fully addresses the health needs of this target group should be able to change and adapt its function over time to accompany people along the retirement path towards healthy ageing. To achieve adaptability, self-learning and the continuous updating of the system seem to be the main challenges to deal with in the present and near future. To meet this challenge, further experiments on large samples should be carried out. It should be underlined that a VC cannot be a substitute for the patient-doctor relationship in the standard health care settings, but needs to achieve comparable trust and loyalty. This demands the future integration of virtual agents into the health care process, as well as trusted recommendations from the health care provider [67,69].

Moreover, it is suggested to carry out such studies in an interdisciplinary way, as recommended by the UCD [42,47], i.e., with the close collaboration of technicians, health care providers, designers, acceptability and usability experts, in order to be able to grasp and interpret all the social and cultural elements that can influence the use of technology in a delicate phase of life such as retirement. At the same time, it can be argued that the long-term use of the virtual coach may potentially lead to a change of beliefs and representations of this system at the socio-cultural and community level. Therefore, those who use it, can be the early beneficiaries of an important change which may lead to a greater collective awareness and to the deconstruction of the stigma around the use of technology in adulthood and later life.

Furthermore, the findings show that cultural and social patterns influenced the end-users’ preferences as regards the functions of the coaching system and the user interface design [38,39,40], and at the same time, provide more details on this topic. The Italian end-users prioritized health prevention functions, whereas the Austrian ranked physical parameters monitoring, and the Dutch rated cognitive capabilities and emotional well-being. The Italian end-users did not stress the function of motivating to physical activity, probably because they were already very active and carried out many outdoor and indoor activities, while this function was stressed by the Austrian and the Dutch participants. Moreover, the Italian end-users found it important to be able to share health goals and achievements and to be connected to other users in order to build up a sense of community belonging. On the other hand, the Austrian and the Dutch end-users preferred not to share their achievements, because they considered health a private business. 

Similarly, the end-users’ preferences on the ECA user interface design changed country to country. For example, the Italian participants did not like the purple background as it recalled unlikely events, e.g., the “sacrifice and death” concepts in the Italian catholic culture, according to which purple is the colour of the Lenten period preceding Easter. This suggests that, when a coaching system is designed, developers need to explore in depth cultural and social patterns throughout the phases of the user centered design process, in order to bring out unconscious cultural and social meanings and transfer them into the functions and design of the system [42]. 

### 4.4. Study Limitations and Suggestions for Future Research

The limited sample, including only white collars, did not allow us to analyse findings by income, educational level, digital literacy, type of work and gender. Moreover, the small number of older workers who agreed to take part in the follow-up telephone interviews during the COVID-19 outbreak did not allow capturing the possible inputs for a personalization of the system in light of the experience of the teleworking and smart-working which older workers could have had during the pandemic. Therefore, it was not possible to highlight the possible influence of the abovementioned factors (i.e., income, educational level, digital literacy, type of work, gender, being in smart-working) on the participants’ attitude towards the use of the system. This suggests that future studies on the use of a VC by older adults involve a wider and multifaceted sample and that adopt a mixed-method approach in order to provide more reliable and generalizable results.

Furthermore, the COVID-19 outbreak made it impossible to carry out the third wave of focus-groups during which end-users could have had the opportunity of testing and interacting with the technology via digital device (e.g., tablet and/or smartphone). Therefore, this limited the full understanding of the end-users’ perspective and the usability of the system. 

## 5. Conclusions

The study shows that the virtual coaching system can be one of the solutions that can best interpret the needs of older workers and retirees of the future, who are expected to be digitalized and long-lived, but at the same time more exposed to chronic diseases and at risk of social isolation due to changes in the organization of work (e.g., tele-working) and relational life (e.g., digital contacts), as consequences of environmental changes and epidemiological emergencies. 

In order to be really effective, the ECA needs to have pleasant appearance and voice, to motivate users without being patronizing nor directive. The main challenge is the design of customizable and not stigmatizing solutions that can adapt to the ageing process and to the evolution of the retirement transition. Currently, big advances are made in robotics, image recognition and speech recognition. Hence, in the near future we can expect more advanced basic technologies, which can be used in more sophisticated virtual coach applications using ECAs. 

Moreover, the study highlighted the potential of the VC functions to address the active and healthy ageing dimensions by greatly expanding the opportunities for accessing, at any time and at low-costs, activities that bring health benefits, e.g., indoor and outdoor physical exercises and mindfulness activities, such that the negative effects of retirement, e.g., sedentary lifestyle, social isolation and lack of stimuli, can be mitigated. 

The current worldwide epidemiological situation has highlighted the urgent need to find innovative solutions to allow older adults to adopt active and healthy ageing-oriented behaviors despite the impossibility of carrying out outdoor activities, meeting friends at home or in a café, and to receive emotional support from family and friends via physical contacts. The COVID-19 outbreak indeed, has triggered a process that could lead to the creation and acceptance of solutions that can be exploited by older adults, even when temporary or unexpected impediments that force them to stay at home, may occur.

## Figures and Tables

**Figure 1 ijerph-18-09681-f001:**
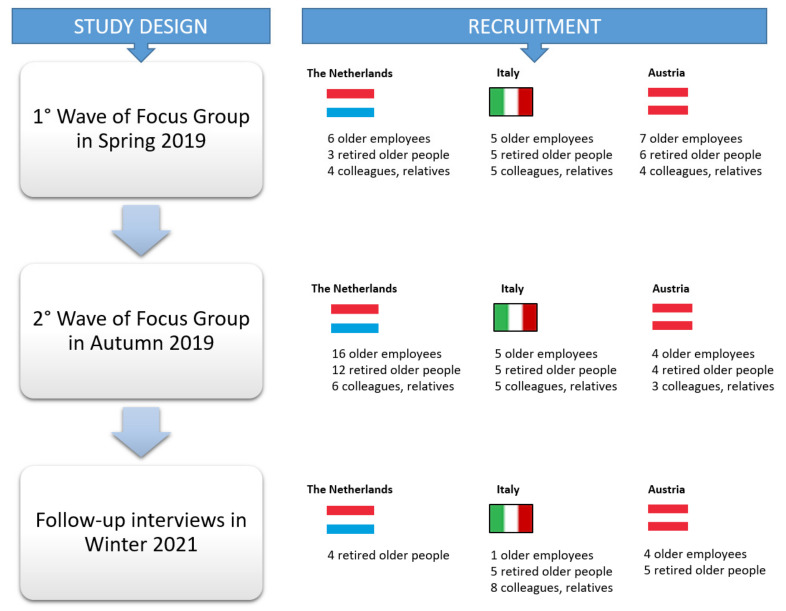
Progress diagram of the data collection phases.

**Figure 2 ijerph-18-09681-f002:**
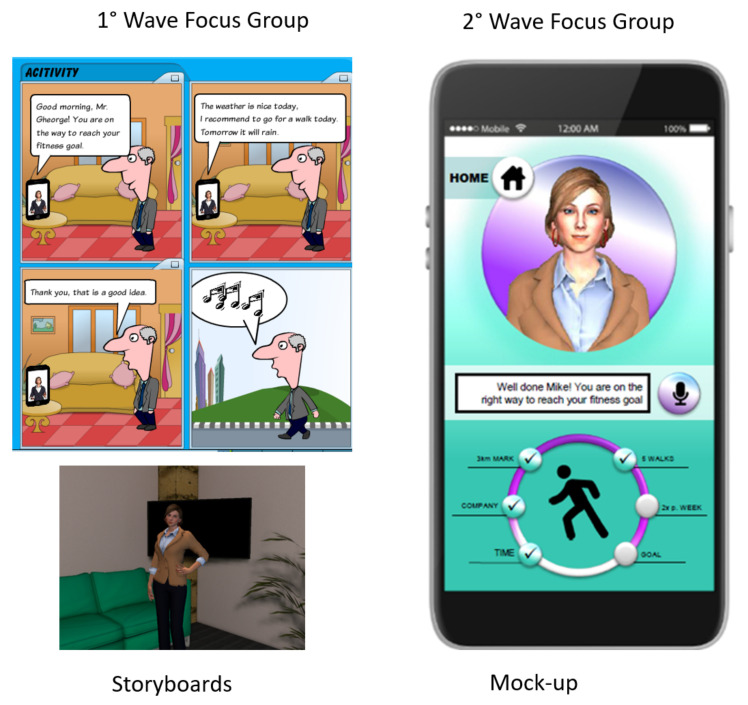
Examples of scenarios, storyboards and mock-ups used during the co-design (first and second focus-group waves).

**Figure 3 ijerph-18-09681-f003:**
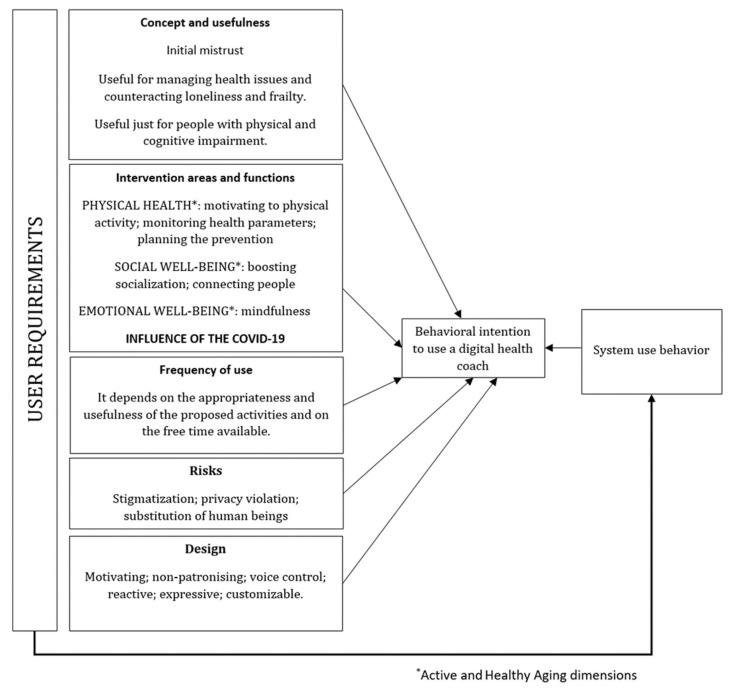
Determinants of behavioral intentions and use behaviors of a VC.

**Table 1 ijerph-18-09681-t001:** Description of participants to the two waves of focus-groups in Austria, Italy and the Netherlands.

**Wave 1**	**Older Workers**	**Retirees**	**Colleagues Relatives**	**Total** **Country**	**Male**	**Female**	**Mean Age**
Austria	7	6	4	17	7	10	59
Italy	5	5	5	15	7	8	59.8
The Netherlands	6	3	4	13	7	6	65.6
Total	18	14	13	45	21	24	61.4
**Wave 2**	**Older Workers**	**Retirees**	**Colleagues Relatives**	**Total** **Country**	**Male**	**Female**	**Mean Age**
Austria	4	4	3	11	7	4	57
Italy	5	5	5	15	7	8	59.8
The Netherlands	16	12	6	34	15	19	64
Total	25	21	14	60	29	31	60.2

**Table 2 ijerph-18-09681-t002:** Participants to the telephone follow-up interviews per country.

Country	Gender		Mean Age	Interviewees			Total
	Male	Female		Retirees	Colleagues of Retirees	Older Workers	
Austria	4	5	60.2	5	0	4	9
Italy	6	8	60	5	8	1	14
The Netherlands	3	1	65.5	4	0	0	4
Total	13	14	61.9	14	8	5	27

**Table 3 ijerph-18-09681-t003:** User requirements: main themes and codes from the focus-groups and the telephone follow-up interviews by country, answering the two Research Questions (RQ).

RQ1 Which Are the User Requirements of a VC Based on an ECA for Training Older Adults in Transition to Retirement and Motivating Them to Adopt Healthy Lifestyle?		RQ2 How Can a VC Mirror the Healthy Ageing Dimensions, Especially during COVID-19 Times?	
User Requirements Themes	Codes	User Requirements Themes	Codes
End-users’ opinion on the VC functions	Encouraging healthy lifestyle (A and NL). Supporting financial planning and concrete activities of retirees (NL). Supporting older people during the transition for managing financial aspects, social activities, health and life style, legal issues (A, IT). Helping people with physical and cognitive limitations (NL). Use of the virtual coach at company level for informing older workers on retirement legal and financial issues (IT). Virtual coach running on portable device (A, IT, NL). Stimulating retirees’ socialization, interests and brain activities (IT).	Physical activity functions	Monitoring end-users’ health (A) Motivating older workers and retirees to practice physical activity (A, IT, NL). Customising physical activities especially for people with special chronic disease (A). Reminding and planning health prevention and screening appointments (IT). Stimulating brain activity (A, NL).
Frequency of use of the VC	Every day use of the virtual coach (A and NL). The use of the virtual coach depends on the free time available, so it is most for retirees (NL). The use of the virtual coach depends on one’s mood (A). The use of the virtual coach depends on users’ family relationships and social condition (IT).	Emotional well-being functions	Helping older workers and retirees have a meaningful life and find purpose in life. Reduce stress and hindering depression (A, IT, NL). Informing users on retirement legislations.
Risks	Stigmatization, privacy violation, substitution of users’ decision making capability (A, IT, NL).	Social relationships functions	Promoting the community construction (IT). Foster retirees’ social relationships (IT). Helping older workers and retirees share and discuss objectives and goals (IT). Promoting the competition among the users for reaching the goals (A, NL).
Design of the ECA	Voice control and setting (A, IT, NL). Reactive (A, IT, NL). Customizable (age, gender, outfit, voice) (A, IT, NL). More expressive and sympathetic (IT). Convincing, motivating and respectful language (i.e., not directive and patronising) (A, IT, NL). Bright and no purple background (IT). Interactive icons giving motivating feedback (e. g. “Well done! You reached your goal!”). Pleasant text field and colours (A, NL).	Useful functions during the COVID-19 outbreak in the 3 realms of health	Encouraging retirees and older workers in smart-working to practice indoor physical and yoga exercises (A, IT, NL). Encouraging retirees re-start going out as soon as possible, according to the restrictions (A, IT, NL). Providing information on movement/social restrictions measures (A, NL). Providing mindfulness exercises, virtual visits, courses and games for emotional well-being (A, IT, NL). Suggesting e-learning opportunities (NL). It might help people ask for social and health support (NL). Creating groups of users based on common interests (e.g., history, bricolage, etc.) (IT).

## Data Availability

Data and materials are available on request from the corresponding author.

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
