# Peer review of "User Requirements Analysis of an Embodied Conversational Agent for Coaching Older Adults to Choose Active and Healthy Ageing Behaviors during the Transition to Retirement: A Cross-National User Centered Design Study"

_ijerph, 2021, doi:10.3390/ijerph18189681_

Round 1

Reviewer 1 Report

Thank you very much for the article submitted for review. I consider the subject and assumptions of the work to be very important. However, I have a few criticisms.

  1. In the material part, there is no description of the research material.
  2. Paragraphs 301 to 319 should be moved to material description.
  3. Tables 1 and 2 for the chapter describing the material. Currently editing of the table is invalid. 
  4. I am sure that a progress diagram should appear
  5. The results chapter reads very badly. It is long and too many general descriptions for this level of the journal. I believe that a list should be prepared, indicating the specific results obtained, indicating the countries and other parameters. Rather in the form of a table or a listing.
  6. Appendix 1 and 2 illegible. Half can not be seen. Perhaps they answer some of my doubts. 

Undoubtedly, the effect of development works is very desirable for the environment of the elderly.

Author Response

Dear Reviewer 1

Thank you for the opportunity to revise our manuscript. We really appreciate the time and effort that you dedicated for providing feedback to our script. The suggestions offered by reviewers have been significant to improve our work and reach a higher level in all aspects of the paper. We addressed each concerns and problems in the main document and point-by-point in this letter, we describe the changes we have made.

We hope the revised manuscript will better suit reviewers and editor and we thank you for the interest in our research.

Sincerely,

The corresponding author

Reviewer 1

Thank you very much for the article submitted for review. I consider the subject and assumptions of the work to be very important. However, I have a few criticisms.

  1. In the material part, there is no description of the research material.

Authors’ answer: in paragraph 2.3. Data collection methodology and tools, the authors described the structure of the two topic-guides of every focus-group wave and included them in the Appendix. The authors also added the following sentence at lines 228-230: “The moderator showed Microsoft Power Point slides to introduce the materials described above”.

  1. Paragraphs 301 to 319 should be moved to material description.

Authors’ answer: Thank you for this suggestion. We would prefer not to move these paragraphs. In fact, paragraphs 301-319 we describe the participants’ characteristics, that often are considered part of the results by this journal. See for example, Santini S, Socci M, D'Amen B, Di Rosa M, Casu G, Hlebec V, Lewis F, Leu A, Hoefman R, Brolin R, Magnusson L, Hanson E. Positive and Negative Impacts of Caring among Adolescents Caring for Grandparents. Results from an Online Survey in Six European Countries and Implications for Future Research, Policy and Practice. Int J Environ Res Public Health. 2020 Sep 10;17(18):6593. doi: 10.3390/ijerph17186593. PMID: 32927827; PMCID: PMC7559354. Available at https://www.ncbi.nlm.nih.gov/pmc/articles/PMC7559354/ and Ng, R. Societal Age Stereotypes in the U.S. and U.K. from a Media Database of 1.1 Billion Words. Int. J. Environ. Res. Public Health. 2021, 18, 8822. https:// doi.org/10.3390/ijerph18168822. Available at file:///C:/Users/santini%20sara/Downloads/ijerph-18-08822%20(2).pdf

  1. Tables 1 and 2 for the chapter describing the material. Currently editing of the table is invalid. 

Authors’ answer: Our apologies for this inconvenient with the tables format whose change after the paper submission was out of authors’ control. The authors amended the table and made them readable.

  1. I am sure that a progress diagram should appear

Authors’ answer: Thank you for this suggestion. The authors added a progress diagram of the data collection phases (new Figure 1).

  1. The results chapter reads very badly. It is long and too many general descriptions for this level of the journal. I believe that a list should be prepared, indicating the specific results obtained, indicating the countries and other parameters. Rather in the form of a table or a listing.

Authors’ answer: Following the Reviewer’s suggestions, the table 1 of Appendix B: User requirements: main themes and codes from the focus-groups and the telephone follow-up interviews answering the two research questions was moved into the Results section because it entails an overall picture of the user requirements also indicating the countries. The table was renumbered Table 3.

In accordance to the amendment described above, the authors added the text in bold to the sentence starting at line 386:

“For every user requirement, several themes were identified in every study country, answering the two study research questions, e.g., “End-users’ feelings on the coaching system concept” or “End-users’ opinions on the coaching system functions”, that are fully reported in Table 3.

Such themes represent the main results of the study that are extensively reported in this section and that might be useful inputs for technology developers to better understand the preferences of older adults in transition to retirement concerning the virtual coaching system and the ECA”

The table 3 follows from line 391.

Since Table 1 was the only table in the Appendix B and it was moved to the Results section, the Appendix B was delated.

Moreover, the authors shortened the text of the Results as much as possible, considering also that the Review n. 2 asked to add more quotations to this section.

  1. Appendix 1 and 2 illegible. Half cannot be seen. Perhaps they answer some of my doubts. 

Authors’ answer: The authors made the tables in the Appendix fully readable. Please, accept our apologies for the formatting issue as explained at point 3.

Undoubtedly, the effect of development works is very desirable for the environment of the elderly.

Reviewer 2 Report

In this article, the authors tried to identify user requirements of Embodied Conversational Agent for coaching older adults. The focus of the study is mostly on motivating older adults to have a healthy lifestyle (ageing well) and how VC can address active and healthy ageing.

The paper is very well-structured and written. As the research is fallen in multi-disciplinary domain, I made some recommendations to enhance the paper clarity from the readers perspective. In these types of articles we need to make sure that there would be no point of confusion based on readers background (either user, technical/developers, physicians, etc. )

 The introduction provides sufficient information but I would recommend that authors identify and define older adults from the geriatric care perspective so that all the readers have the same understanding from the age group (for example are you referring to age 65+ - check Tun, S.Y.Y.; Madanian, S.; Parry, D. Clinical Perspective on Internet of Things Applications for Care of the Elderly. Electronics 20209, 1925. https://doi.org/10.3390/electronics9111925 and Encyclopedia of Gerontology and Population Aging)

Also, providing some background information about embodied conversational agents help readers to have a better understanding of the technology and the possible differences between similar technologies and wearable devices.

it also worth adding a background of the research so that readers get familiar with the background and research gap. This would enhance the explanation of the significance of the research. 

The methodology is appropriate and discussed well with enough details. 

Although the result section is good in terms of quality and content, I would recommend including more direct quotes from the study participants. As the research used qualitative methodology and focus groups, sharing more participants feelings, demands and requirements would be quite helpful to support the statements and discussion. Also, they might be useful and technology developers to have an understanding of user preference and demand. 

Author Response

Dear Reviewer 1

Thank you for the opportunity to revise our manuscript. We really appreciate the time and effort that you dedicated for providing feedback to our script. The suggestions offered by reviewers have been significant to improve our work and reach a higher level in all aspects of the paper. We addressed each concerns and problems in the main document and point-by-point in this letter, we describe the changes we have made.

We hope the revised manuscript will better suit reviewers and editor and we thank you for the interest in our research.

Sincerely,

The corresponding author

Reviewer 2

In this article, the authors tried to identify user requirements of Embodied Conversational Agent for coaching older adults. The focus of the study is mostly on motivating older adults to have a healthy lifestyle (ageing well) and how VC can address active and healthy ageing.

The paper is very well-structured and written. As the research is fallen in multi-disciplinary domain, I made some recommendations to enhance the paper clarity from the readers perspective. In these types of articles we need to make sure that there would be no point of confusion based on readers background (either user, technical/developers, physicians, etc. )

  1. The introduction provides sufficient information but I would recommend that authors identify and define older adults from the geriatric care perspective so that all the readers have the same understanding from the age group (for example are you referring to age 65+ - check Tun, S.Y.Y.; Madanian, S.; Parry, D. Clinical Perspective on Internet of Things Applications for Care of the Elderly. Electronics20209, 1925. https://doi.org/10.3390/electronics9111925 and Encyclopedia of Gerontology and Population Aging)

Authors’ answer: To address the Reviewer’s suggestion, the authors clearly defined the age range of the older adults who the system is targeted to i.e. older adults aged 55 and over.

Nevertheless, there is not consistency in the literature about the age range and the definition of older adults. In general, what we noticed is that the group of older adults is a general expression for defining people over 55 years but without clinical specificities. Thus, we cited:

Petry NM. A comparison of young, middle-aged, and older adult treatment-seeking pathological gamblers. Gerontologist. 2002 Feb;42(1):92-9. doi: 10.1093/geront/42.1.92. PMID: 11815703

Tun, S.Y.Y.; Madanian, S.; Parry, D. Clinical Perspective on Internet of Things Applications for Care of the Elderly. Electronics 2020, 9, 1925.

and we rephrased the sentence beginning at line 70 as follows (in bold the added text and references):

One of such changes is represented by the transition from work to retirement, which generally concerns people aged between 55 and 67 years according to the different pension systems and regulations that vary from country to country. This group of older adults also is in transition from the adulthood to the old age and so they do not fully fall in the traditional demographic and geriatric age and/or care classifications [18, 19]. In fact, what defines them are not only age or long-term care needs, but even a multiple set of dimensions among which the level of participation in the society and the position in the labour market, identified from time to time by every study”.

To further clarify the virtual coaching system target we also added the following details (in bold) in the sentence starting at line 145:

“This study aims to translate the active and healthy ageing dimensions into a virtual coaching system supporting over 55 people, without long-term care needs, in aging well and healthily during their transition from work to retirement”.

  1. Also, providing some background information about embodied conversational agents help readers to have a better understanding of the technology and the possible differences between similar technologies and wearable devices.

Authors’ answer: To address the Reviewer’s suggestion, the authors added the following text. “Indeed, ECAs differ from other technologies (e.g., wearable devices) for having the same proprieties as humans involved in conversations such as the capacities to exchange verbal and non-verbal communication. The unprecedented advantage of having such rich style of communication and interaction modality lies in the fact that it offers human-like speech, facial expressions, hand gestures, and body stance and it is supportive at any time and in any location owing to the use of applications on smartphones or tablets.”

  1. it also worth adding a background of the research so that readers get familiar with the background and research gap. This would enhance the explanation of the significance of the research. 

Authors’ answer: The authors added the following sentence in the Introduction: “The first novelty of the study lies in considering a target that is often overlooked by research on the development of this kind of technical solution: people aged between 55 and 70 years, in transition from work to retirement. Furthermore, a new contribution for transferring the healthy ageing paradigm and the life course perspective to a virtual coaching system. It is also one of the first studies that monitored the end-users’ change of perspective concerning a virtual coach and an ECA throughout the Covid-19 outbreak. Finally, the collection of the end-users’ inputs on the ECA had enriched the knowledge on this innovative solution and paved the way towards its full personalization.”

  1. The methodology is appropriate and discussed well with enough details. 

Authors’ answer: Thank you for your appreciation.

  1. Although the result section is good in terms of quality and content, I would recommend including more direct quotes from the study participants. As the research used qualitative methodology and focus groups, sharing more participants feelings, demands and requirements would be quite helpful to support the statements and discussion.

Authors’ answer: The authors added just some and the most meaningful quotations, in order to also address the request of the Reviewer n. 1 to reduce the Results text length.

In addition, the authors harmonized the format of the quotations (now all in Italics) and the infos in brackets after the quotations.

  1. Also, they might be useful and technology developers to have an understanding of user preference and demand. 

Authors’ answer: Table 1 in the Appendix B was moved to the Results for clearly giving an overview of the user requirements and a sentence was added at line 403 to explain that the inputs in the table would be helpful for developers. In bold the added text.

“For every user requirement, several themes were identified in every study country, answering the two study research questions […]. Such themes might be useful inputs for technology developers to better understand the preferences of older adults in transition to retirement concerning the virtual coaching system”.

Furthermore, we revised the format of the tables to make them fully readable. 

Finally, we rise your attention to the addition of the Figure 1 that meets a request of the Reviewer n. 1.

Round 2

Reviewer 1 Report

Thank you for introducing the corrections I have proposed. I accept work in this version.

This manuscript is a resubmission of an earlier submission. The following is a list of the peer review reports and author responses from that submission.